# Icariin Alleviates Nonalcoholic Fatty Liver Disease in Polycystic Ovary Syndrome by Improving Liver Fatty Acid Oxidation and Inhibiting Lipid Accumulation

**DOI:** 10.3390/molecules28020517

**Published:** 2023-01-05

**Authors:** Yang Hai, Ling Zuo, Meng Wang, Ruoyu Zhang, Munan Wang, Li Ren, Congwen Yang, Jianwei Wang

**Affiliations:** 1Traditional Chinese Medicine Key Laboratory in Chongqing for Prevention and Cure of Metabolic Diseases, Chongqing 400016, China; 2College of Basic Medicine, Chongqing Medical University, Chongqing 400016, China; 3College of Traditional Chinese Medicine, Chongqing Medical University, Chongqing 400016, China

**Keywords:** fatty acid oxidation, icariin, liver lipid deposition, nonalcoholic fatty liver disease, polycystic ovary syndrome

## Abstract

(1) Background: Icariin is the main component of the Chinese herb Epimedium. A number of studies have shown that it alleviates abnormal lipid metabolism. However, it is not clear whether and how icariin can ameliorate hepatic steatosis with polycystic ovary syndrome (PCOS). This study was designed to explore the anti-hepatosteatosis effect of icariin in rats with polycystic ovary syndrome. (2) Methods: Female Sprague Dawley(SD)rats were treated with a high-fat diet and letrozole for 21 days to make nonalcoholic fatty liver disease (NAFLD) in the polycystic ovary syndrome model. Then model rats were treated with icariin (by gavage, once daily) for 28 days. Serum hormones and biochemical variables were determined by ELISA or enzyme. RNA-sequence analysis was used to enrich related target pathways. Then, quantitative Real-time PCR (qRT-PCR) and Western blot were performed to verify target genes and proteins. (3) Results: Icariin treatment reduced excess serum levels of Testosterone (T), Estradiol (E2), Luteinizing hormone (LH), Follicle-stimulating hormone (FSH), LH/FSH ratio, insulin, triglycerides (TG), and aspartate aminotransferase (AST) in high-fat diet (HFD) and letrozole fed rats. Meanwhile, icariin ameliorated HFD and letrozole-induced fatty liver, as evidenced by a reduction in excess triglyceride accumulation, vacuolization, and Oil Red O staining area in the liver of model rats. Results of RNA-sequencing, western blotting, and qRT-PCR analyses indicated that icariin up-regulated fatty acid translocase (CD36), in mitochondria, and peroxisome proliferator-activated receptor α (PPARα) expression, which led to the enhancement of fatty acid oxidation molecules, such as cytochrome P450, family 4, subfamily a, polypeptide 3 (CYP4A3), carnitine palmitoyltransferase 1 α (CPT1α), acyl-CoA oxidase 1 (ACOX1), medium-chain acyl-CoA dehydrogenase (MCAD), and long-chain acyl-CoA dehydrogenase (LCAD). Besides, icariin reduced lipid synthesis, which elicited stearoyl-Coenzyme A desaturase 1 (SCD1), fatty acid synthase (FASN), and acetyl-CoA (ACC). (4) Conclusion: Icariin showed an ameliorative effect on hepatic steatosis induced by HFD and letrozole, which was associated with improved fatty acid oxidation and reduced lipid accumulation in the liver.

## 1. Introduction

An estimated 5% to 20% of women of reproductive age suffer from polycystic ovary syndrome and endocrine disorders [1]. The disorder usually manifests as hyperandrogenemia, irregular menstrual cycles, abnormal ovarian function, polycystic follicles (polycystic ovary syndrome), and insulin resistance (IR) [2,3,4]. Histologically, non-alcoholic fatty liver disease (NAFLD) is similar to alcoholic liver disease in terms of hepatic fat accumulation, in which patients do not drink alcohol or drink small amounts of alcohol [5,6]. The pathogenesis of NAFLD is multifactorial, but obesity and IR seem to be the key causative factors. As far as histology is concerned, it ranges from simple steatosis (SS), where fat accumulates in the liver, to non-alcoholic steatohepatitis (NASH) [7]. There is growing evidence that the incidence of non-alcoholic fatty liver disease increases when patients already have polycystic ovary syndrome. The prevalence of NAFLD has been reported to be higher in patients with Polycystic Ovary Syndrome (PCOS) (51%) than in women without the syndrome (34%) [8]. Several studies using laboratory assessments to diagnose non-alcoholic fatty liver disease in patients with PCOS have shown a strong association with androgens [1,9,10]. Further research is urgently needed in the area of NAFLD in PCOS, as there is no definitive pharmacological treatment.

NAFLD is characterized by an increased accumulation of lipids in the liver, which may be caused by a number of factors [11]. Deficiencies in lipid disposal and acquisition account for fat accumulation in the liver. Four main pathways regulate them: uptake of circulating lipids, de novo lipogenesis (DNL), fatty acid oxidation (FAO), and export of lipids from very low-density lipoproteins (VLDL). Numerous studies in rodents have attempted to indirectly promote FAO to reduce systemic obesity and prevent the development of metabolic disorders associated with obesity [12,13]. The liver is largely dependent on fatty acid transporters for the uptake of circulating fatty acids, with a smaller contribution from passive diffusion [14]. It is mediated by fatty acid transport protein (FATP), and cluster differentiation 36 (CD36) [15]. Miquilena-Colina M E’s study suggested that the translocation of CD36 protein from the cytoplasm to the cell membrane may promote the development of NAFLD [16]. Liver fatty acid oxidation occurs mainly in mitochondria, and CD36 can regulate the expression of peroxisome proliferator activated receptor α (PPARα) factors that control FAO [15]. By activating PPARα, FAO-related genes are expressed in the mitochondria, peroxisomes, and cytochromes, thereby reducing liver lipid levels [17].

Traditional Chinese medicine has been proven to be a great potential in the treatment of metabolic diseases. Epimedium brevicornu Maxim (Family Berberidaceae), has important medicinal properties for treating several ailments, including several metabolic disorders and liver cancer. Icariin, also known as Xianling spleen, is one of the left and right compounded herbs in Guiyangtang Junmai, from the Shen Nong Ben Cao Jing. Its main effects are to tonify the kidney and strengthen the yang, dispel wind, and remove dampness. Studies have shown that icariin contains active ingredients, which have estrogen-like effects. In addition, phytoestrogen-like active substances can promote the secretion of sex hormones and regulate the reproductive endocrine system. Besides, icariin has a variety of biological functions. It can regulate lipid metabolism and also alleviate various metabolic diseases caused by abnormal lipid metabolism, such as obesity and diabetes [18,19,20]. Icariin is a naturally occurring phytochemical agent primarily extracted from Epimedium. It has been mentioned in the literature that icariin has strong anti-inflammatory effects [21,22]. However, it is not clear whether and how icariin can improve non-alcoholic fatty liver disease in PCOS.

In this study, we evaluated whether icariin could alleviate NAFLD in PCOS by modulating lipid metabolism and the underlying molecular mechanisms. Letrozole is an aromatase inhibitor that is reported to affect testosterone accumulation and upregulates liver triglyceride levels after 21 days of administration. Therefore, we used 1 mg/kg of letrozole and HFD co-administration to construct the rat models [23]. Our results showed that icariin not only improved sex hormone levels but also reduced lipid deposition in the liver and improved the oxidative function of mitochondria. This study may prove to be a new strategy in the prevention of non-alcoholic fatty liver disease in polycystic ovary syndrome by using traditional Chinese medicine monomeric icariin.

## 2. Results

### 2.1. Rat Model and Sex Hormone Levels

Before the treatment, vaginal exfoliated cells in the vaginal smears of 36 female rats during normal estrus, were observed microscopically for seven consecutive days (Figure 1A). After 21 days of letrozole gavage, 27 rats in the model group showed endocrine and estrous cycle disorders. Observation of vaginal smear cell morphology showed all cells to be in the diestrus phase. The ELISA results showed a significant decrease in serum E2 levels and an increase in serum T levels in the model group (Figure 1B), indicating that the rat model of polycystic ovary syndrome had been successfully established. After 28 days of administration, ELISA kits detected the rat’s sex hormones. Plasma levels of T, E2, LH, FSH, LH/FSH, fasting blood glucose, INS, and HOMA-IR were shown to have improved significantly (Figure 1C). According to our findings, HOMA-IR, fasting blood glucose, INS, LH, T, and LH/FSH dropped by 25.6%, 8.9%, 19.0%, 13.6%, 23%, and 33.8%, respectively, with high-dose icariin treatment compared to low-dose icariin treatment. In comparison to low-dose drug therapy, E2 and FSH in the high-dose drug treatment increased by 16.3% and 23.7%.

Moreover, it was evident that body weight gain and liver weight in the model group were higher than those in the control group and clearly decreased after treatment with icariin (*p* < 0.05). The liver weight of the high-dose drug group decreased by 5.8%, and the liver weight, as a percentage of total weight, decreased by 10.5%. Among the groups of rats, there was no significant difference in feeding volume (Figure 1D).

### 2.2. Icariin Improves Liver Steatosis and Liver Function in Rats

We looked at the hepatic lipid profile in order to learn how icariin affected the rat liver steatosis by HFD and letrozole. As expected, serum TG and liver TG in the model rats were significantly higher than those in the control group. This suggests that lipid deposition occurred in the liver of the polycystic ovary syndrome rat model. However, the icariin treatment reversed the increase in TG content induced by HFD and letrozole, while the icariin treatment showed a decreasing trend. Meanwhile, we found no significant difference in TC among the four groups (Figure 2B). Similarly, HE (hematoxylin-eosin) and Oil Red O staining showed significant lipid deposition in the rat liver, disorganized hepatocyte structure, drops of different sizes, and changes in the model group (Figure 2A). The effect of high-dose icariin treatment is more apparent, which is in line with earlier findings. The Oil Red O stained area, serum TG, and liver TG all decreased in comparison to low-dose icariin by 2.04%, 16.2%, and 23.7%, respectively.

Liver function was assessed by assays for serum ALT, AST, HDL-C, and LDL-C. As shown in Figure 2B, icariin could ameliorate the liver function impairment caused by HFD and letrozole to some extent, although there was no statistical significance. So, these results demonstrated that icariin showed an evident protection against HFD and letrozole-induced hepatic steatosis.

### 2.3. Effects of Icariin on NAFLD in PCOS Models as Revealed by RNA-Sequence Analysis

To further analyze the mechanism of how icariin improved liver steatosis in PCOS, we performed the analysis by RNA-Sequencing. The differential genes of the two groups were visualized, and the stacked charts of the differential gene and the Wayne diagram were plotted. As shown in the Figure 3, there were a total of 594 and 321 differential genes, respectively. There were 197 (61%) up-regulated genes and 124 (39%) down-regulated genes in the control and model groups, 285 (48%) up-regulated genes, and 309 (52%) down-regulated genes in the Model and H-ICA groups. A total of 121 genes observed a significant and differentially expressed expression pattern following treatment with HFD-letrozole and icariin among three groups named co-regulatory genes (Figure 3A,B). Principal Component Analysis (PCA) analysis showed that three groups of rats were completely separated (Figure 3C).

Then we performed a Kyoto Encyclopedia of Genes and Genomes (KEGG) pathway analysis on 121 differently expressed genes (Figure 3D). There were many differentially expressed genes enriched in fatty acid degradation pathways. Therefore, we further analyzed the differential genes with higher expression amounts and found that HFD and letrozole-induction significantly decreased CYP4A3 expression (cytochrome P450, family 4, subfamily a, polypeptide 3), whereas icariin treatment markedly increased its expression in the rat liver (Figure 3E). qRT-PCR and Western-blot analysis obtained similar observations (Figure 3F).

### 2.4. Icariin Improves NAFLD in PCOS by Promoting Fatty Acids Oxidation in the Liver

It is reported that CYP4A3 was the target gene for PPARα (Peroxisome proliferator-activated receptor α) and PPARα was able to affect liver mitochondrial fatty acid oxidation [24,25]. Additionally, FAT/CD36 (fatty acid translocase) could affect the expression of PPARs (peroxisome proliferators-activated receptors) [26]. Our previous studies have demonstrated that CD36 can improve oxidative function by up- regulating PPARα expression, thereby reducing lipid deposition [27,28]. Based on this evidence, we wanted to know if icariin therapy activated the CD36-PPARα signaling pathway. We examined the impact of icariin on CD36-PPARα in the rat liver to test this theory. In line with expectations, icariin therapy raised the mRNA and protein expression of PPARα as well as the mRNA levels of CPT1α, ACOX1, MCAD, and LCAD (Figure 4A,B).

In addition, we also measured the levels of fatty acid transport protein 2 (FATP2) and fatty acid binding protein 1(FABP1), two critical enzymes for lipid accumulation in the rat liver [29]. Importantly, icariin treatment can greatly down-regulate the increase in FATP2 and FABP1 caused by HFD and letrozole (Figure 4A,B). These results suggest that icariin promoted liver fatty acid oxidation to improve non-alcoholic fatty liver disease in polycystic ovary syndrome. However, we noticed that the mRNA and protein expression of CD36 was contrary to our expectations, which caused us to consider it.

### 2.5. Icariin Alleviates NAFLD in PCOS by Increasing CD36 Content in Mitochondria

Fatty acid transporter CD36 (CD36) is a membrane glycoprotein that facilitates long-chain fatty acid (LCFA) uptake and intracellular transport [30]. As the function of CD36 is related to its subcellular localization [31]. So, we speculated whether CD36 had undergone translocation, resulting in the opposite expression pattern than the one expected. To test our theory, we looked at the cytoplasm, mitochondria, and membrane expression of CD36. Icariin administration reversed the HFD and letrozole-induced a rise in CD36 expression level in the membrane, as demonstrated in Figure 5A. In the mitochondria, icariin altered the low expression of CD36 caused by HFD and letrozole. However, in the cytoplasm, icariin treatment did not cause appreciable changes in CD36 expression.

Then, immunofluorescence of the liver showed that CD36 signaling was localized in the mitochondria of the liver (Figure 5B). The CD36 fluorescence intensity in the model group was lower than in the blank group, while in the drug group, this phenomenon improved. The result suggested that icariin can reduce the accumulation of liver fatty acids. On the one hand, icariin reduced the expression of the liver membrane protein CD36. On the other hand, it also increased the expression of CD36 mitochondria. However, both have been found to enhance fatty acid oxidation of the liver and reduce lipid deposition, in order to improve non-alcoholic fatty liver disease in polycystic ovary syndrome.

### 2.6. Icariin Improves Liver Steatosis by Reducing the Expression of Genes Associated with Fatty Acid Synthesis

Since lipid over-synthesis is also an important factor in the formation of non-alcoholic fatty liver [17,32], we verified the expression of icariin in liver fatty acid synthesis-related genes.

The essential trans-activators of fatty acid production in the liver are ChREBP (carbohydrate response element-binding protein) and SREBP1c (sterol regulatory element-binding protein 1c), which are closely related to hepatic steatosis induced by HFD [33,34]. Excessive consumption of HFD resulted in the activation of ChREBP and SREBP1c transcripts, increasing the expression of SCD1 (stearoyl-Coenzyme A desaturase 1), FASN (fatty acid synthase), ACC (acetyl-CoA carboxylase), and LXRα (nuclear receptor subfamily 1, group H, member 3), thus altering the composition of fat in the rat liver. Therefore, we explored whether icariin down-regulated the expression of SCD1, FASN, ACC, and LXRα by inhibiting the transcription of SREBP1c and ChREBP.

As shown in Figure 6, icariin was observed to significantly inhibit the expression of fatty acid synthesis at the mRNA level, including FASN, SCD1, and ACC, although its ameliorating effect on SREBP1c, ChREBP, and LXRα was not noticeable. These results suggested that icariin can reduce the expression of genes related to fatty acid synthesis in the liver, alleviating liver steatosis to some extent.

## 3. Materials and Methods

### 3.1. Experimental Animals, Drugs, and Reagents

A total of 36 Specific Pathogen Free (SPF)-rated 4–6-week-old female Sprague Dawley (SD) rats were purchased from the Animal Experiment Center of Chongqing Medical University. This study was approved by the Animal Experiments Ethical Review Committee of the Chongqing Medical University, Chongqing, China. The reference number is 2021027. The rats were routinely housed at a laboratory temperature (21 ± 1) °C, with a relative humidity of 55% ± 5%, with alternating light and dark conditions for 12 h, and drinking water, which they could drink ad libitum. After 7 days of adaptive rearing at the SPF level, rats were randomly divided into a control group (1% CMC 1 mg/kg, *n* = 9) and a model group (letrozole 1 mg/kg, *n* = 27). The control group was given a normal diet and the model group was given a high fat diet (D12492, Research Diets, New Brunswick, NJ, USA. Rodent Diet With 60% kcal Fat. Ingredients: Protein: 203.0 g, Carbohydrate: 197.8 g, Fiber: 50 g, Fat: 270.0 g, Mineral: 50.0 g, Vitamin: 3.0 g, Dye: 0.05 g. Total: 773.85 g), all drank water freely. Food intake was recorded every 2 days and body weight was recorded every 3 days. At the end of the 3rd week, blood was drawn to measure the serum levels of sex hormones in the rats. When the estrous cycle and hormonal profiles were changed, the polycystic ovary syndrome model was successfully established. Rats were selected according to their estrous cycle and sex hormone results, then they were divided into the control group (1% CMC 1 mg/kg, *n* = 9), model group (1% CMC 1 mg/kg, *n* = 9), icariin low dose group (L-ICA, 40 mg/kg, *n* = 9), and icariin high dose group (H-ICA, 80 mg/kg, *n* = 9). All groups were given a normal diet and water, and the food intake recorded every 2 days and body weight recorded every 3 days. In the icariin treatment group, the samples were dissolved in 1% hydroxyethylcellulose solution (CMC) for gastric lavage. Gastric lavage was performed once a day for 4 weeks. The rats underwent isoflutane anesthesia at the conclusion of the procedure, and blood was drawn from the abdominal aorta. Livers were weighed and divided into two parts, one for histological staining and the other for molecular expression assays.

Letrozole (2.5 mg/piece, Lianyungang, China) was acquired from Jiangsu Hengrui Pharmaceutical Co., Ltd. (Lianyungang, China) Icariin was purchased from Chengdu Croma Biotechnology Co., Ltd. (Chengdu, China). A 1% CMC (CAT: 0714A22) was obtained from LEAGUE. Cytoplasmic and cell membrane protein extraction kits (CAT: SM-005) were purchased from Invent Biotechnologies (2605 Fernbrook Ln N Ste A, Plymouth, MN 55447, USA). Mitochondrial Extraction Kit (CAT: SM0020) was purchased from Solarbio Science and Tehnology Co., Ltd. (Beijing, China).

Antibodies against CD36 (CAT: NB400-144), CPT1α (CAT: ab128568), PPARα (CAT: ab8934), CYP4A3 (CAT: ab140635), Na/K ATP (CAT: ab7671), and VDAC1 (CAT: ab14734) were purchased from Abcam (Cambridge, UK). FATP2 (CAT: 14048-1-AP) was obtained from Proteintech (Proteintech Group, Inc. 5500 Pearl Street, Ste 400, Rosemont, IL 60018, USA). FABP1 (CAT: PTM-6412) was from PTM Bio (Hangzhou, China), and β-actin (CAT: 4970) was purchased from Cell Signaling Technology (Beverly, MA, USA) (Appendix A).

### 3.2. Estrous Cycle Determination

A vaginal swab was taken after modeling and during the fourth week of icariin treatment. The swab was taken at the same time every day, for seven days. Microscopic (BX53, Olympus Corporation, Tokyo, Japan) analysis of the predominant cell type in vaginal smears using 0.5% Methylene Blue Solution (CAT: #G1302; Solarbio Science and Technology Co., Ltd., Beijing, China) confirmed the estrous stage.

### 3.3. Plasma Sex Hormones, Fasting Blood Glucose and the Homeostasis Model Assessment of the IR Index

Plasma was obtained from blood samples by centrifuging them at 3500 rpm for 20 min. Serum levels of Estradiol (E2), Testosterone (T), Follicle-stimulating hormone (FSH), Luteinizing hormone (LH), LH/FSH ratio, and Fasting insulin were measured using ELISA kits, in accordance with the manufacturer’s instructions. Homeostasis model assessment of insulin resistance (HOMA-IR) was calculated as follows: (fasting blood glucose × serum insulin)/22.5. At least six rats/group were used for each assay.

### 3.4. Measurement of Serum and Liver Biochemical Markers

Using biochemical assay kits were used in complete compliance with the directions in the kit handbook (Nanjing Jiancheng Institute of Biological Engineering, Nanjing, China). High-density lipoprotein cholesterol (HDL-C), low-density lipoprotein cholesterol (LDL-C), serum triglycerides (TG), total cholesterol (TC), and low-density lipoprotein cholesterol (LDL), and the enzymes alanine aminotransferase (ALT) and aspartate aminotransferase (AST) were measured. An enzymatic assay was performed.

Rats’ TG and TC levels in their livers were also measured. Briefly stated, the liver tissues were weighed and recorded correctly before being extracted with isopropanol at a ratio of 50:1. (50 mg of tissue dissolved in 1 mL of isopropanol). The tissue was then suitably homogenized by the addition of grinding beads, and the homogenate was then placed on a four-dimensional rotary mixer blender overnight at 4 °C (Beyotime, Shanghai, China). After centrifuging the homogenate of the liver tissue at 3000 rpm for 10 min, the amounts of TG and TC in the supernatant were measured using a commercial test kit (Jiancheng, Nanjing, China). The manufacturer expressed the TG by dividing the TG (or TC) values by the weight of the liver tissue.

### 3.5. Histopathology

A portion of the liver was treated with 10% formalin and encapsulated in paraffin. 4 μm sections were cut and stained with hematoxylin and eosin (HE) to examine the histology of the liver.

After fixing with 4% paraformaldehyde, the dehydrated liver tissues were twice submerged in a 15–30% sucrose solution at 4 °C. In the following steps, the tissue samples were embedded using embedding agents. Next, they were dyed with Oil Red O (ORO) after being chopped into 8-mm chunks, using an automatic slicer. Finally, the sections were stained with hematoxylin, then washed 3 times with pure water and sealed with glycerol gelatin. The liver lipid droplets were observed microscopically. Measurements were made using Image J software 8.0.2 (2365 Northside Dr. Suite 560, San Diego, CA 92108, USA) and the ratio of Oil Red O staining area to total tissue area (%) was calculated.

### 3.6. Gene Expression Differences Are Identified through RNA-Sequence Analysis

Using the TRIzol^®^ reagent and DNase I (TaKara), total RNA was extracted from rat livers (3 samples per group, comprising the control, model and drug groups), in accordance with the manufacturer’s instructions (Invitrogen, Carlsbad, CA, USA). The quality of the RNA was assessed using an Agilent 2100 Bioanalyzer. Sequencing libraries were prepared based on poly A-rich mRNA in biological replicates. RNA-seq libraries were prepared from poly(A)-rich mRNA as described previously [35]. The library was size-selected for a 300 bp cDNA target fragment on a 2% low range super agarose. After quantification with TBS380, the paired-end RNA-seq sequencing libraries were sequenced (2 × 150 bp read length) using an Illumina HiSeq Xten/NovaSeq 6000 sequencer.

The RNA-seq study was carried out as previously explained [35]. Transcripts per million readings (TPM) were used to identify the genes that showed differential expression (DEGs) between the control, model, and treatment groups. To identify DEGs, *p*-values 0.05 was used as cutoffs for up- and down-regulated genes. Additionally, using the Gene Ontology (GO) and Kyoto Encyclopedia of Genes and Genomes (KEGG) annotation systems, the functional enrichment of genes with varying levels of expression was estimated. In the end, genes of interest and pathways for enrichment were selected for further study.

### 3.7. Quantitative Real-Time PCR

Total RNA was extracted from 50 mg of rat liver tissue using AG RNAex Pro Reagent (code: AG 21102), and the reagents were used according to the reagent instructions. Evo M-MLV RT Premix for qPCR (code: AG11706), a reverse transcription reagent, was used to convert total RNA into cDNA for the qRT-PCR experiment, in accordance with the SYBR^®^ Green Premix Pro Taq HS qPCR kit (code: AG11701). Each sample’s mRNA expression underwent repeated examination and normalization to glyceraldehyde-3-phosphate dehydrogenase (GAPDH) mRNA. Cycling conditions were as follows: initial denaturation at 95 °C for 30 s, followed by cycling at 94 °C for 5 s, 60 °C for 15 s, and 72 °C for 10 s (CFX Connect, Bio-Rad, 1000 Alfred Nobel Drive, Hercules, CA 94547, USA). The collected data was examined using the 2−ΔΔCt technique. Primers for qRT-PCR on Table 1.

### 3.8. Western Blot

Briefly, liver tissues were dissected and lysed in RIPA lysate (CAT: P0013B) and centrifuged, and the supernatant collected. Bicinchoninic acid (BCA) protein concentration assay kit (CAT: P0012A) was used to determine protein concentration. Vertical electrophoresis separation was performed. The primary antibody incubated with a membrane at 4 °C, for 12–16 h. TBST (TBS + Tween) was washed 3 times for 15 min each. Then, the secondary antibody was incubated with a membrane at room temperature, for 2 h. After 3 washes of TBST, the membrane was irradiated with a chemiluminescent solution. The developed strips were scanned with Odyssey Fc (Odyssey Fc, LI-COR Biosciences, 4647 Superior Street, Lincoln, NE, USA) two-color infrared fluorescence imaging system, and the gray value of each strip was determined by analysis with Image J software. The ratio of the gray value of the obtained target strip to the gray value of the corresponding internal reference β-actin strip was used to determine the relative expression of the corresponding target protein.

### 3.9. Data and Statistical Analysis

The statistical analysis was done with the help of GraphPad Prism 8.0. All data were compared as mean ± SEM, and *p* < 0.05 was regarded as statistically significant. ANOVA was used for comparisons across several groups, and the *t*-test was employed for comparison of the two groups.

## 4. Discussion

Polycystic ovary syndrome is an endocrine disorder with a reported global prevalence of 6–18%. The main pathological features include elevated androgen levels, insulin resistance, dyslipidemia, and a chronic low degree of inflammation [2,3]. Non-alcoholic fatty liver disease (NAFLD) is considered to be a hepatic manifestation of metabolic syndrome (MetS). Non-alcoholic fatty liver disease has been defined as fatty liver in patients without alcohol intake and in patients excluded from other diseases. In the common population, the prevalence of the non-alcoholic fatty liver disease is 25% [36]. In recent years, with the increasing number of clinical cases of non-alcoholic fatty liver disease in PCOS, many researchers are focusing on the relationship between the two. Women with polycystic ovary syndrome have been found to have higher rates of NAFLD and NASH diagnosed by liver biopsy, magnetic resonance spectroscopy, computed tomography, and abdominal ultrasound [37,38]. A cross-sectional study using ultrasound in a Chinese population showed a 56% prevalence of NALFD in patients with PCOS, but only 38% in healthy women [39,40]. These results indicate that the prevalence of NALFD increases progressively with the advent of polycystic ovary syndrome. Unfortunately, no definitive drug has been found to prevent PCOS from becoming alcoholic fatty liver.

In Asia, traditional Chinese medicine (TCM) has been widely used to treat various metabolic diseases, and inflammatory diseases, including non-alcoholic fatty liver disease and polycystic ovary syndrome. The holistic concept and staged treatment of NAFLD in TCM have shown their advantages in the treatment of this complex metabolic disease [2,20]. TCM monomeric compounds have a defined molecular formula and spatial structure that makes them active substances. Currently, several herbal monomers have been shown to have hepatoprotective, anti-inflammatory, and anti-oxidant effects, such as resveratrol, micronutrients, and curcumin [41]. In the present study, the NAFLD in a PCOS rat model was used as the subject of discussion and different concentrations of the herbal monomer icariin were used to treat the rats. The results clearly showed that icariin significantly improved hepatic lipid deposition in rats with polycystic ovary syndrome.

Icariin improves hepatic steatosis in PCOS rats via the CD36-PPARα pathway. We examined TC, TG, ALT, AST, HDL-C, and LDL-C in the liver and serum of rats. The current study was in line with our earlier research and other related studies, which found that levels of TC, HDL, LDL, and AST were not significantly affected by HFD feeding [42,43]. Importantly, we saw that rats treated with icariin had considerably lower levels of serum TG, ALT, and liver TG. These results indicated that liver function was impaired in polycystic ovary syndrome rats, while icariin treatment improved the HFD and letrozole-induced liver dysfunction. HE staining and Oil Red O staining further validated our results. All the evidence indicated that icariin alleviated the liver lipid deposition in polycystic ovary syndrome rats. To clarify the underlying mechanism, we explored transcriptomics. We ranked the relative expressions of 121 differential genes in descending order, and finally screened a molecule closely related to fatty acid oxidation, CYP4A3, for subsequent validation. Then, CYP4A3 was identified and validated at the gene and protein levels. The results demonstrated that icariin treatment significantly increased its expression in the rat liver. It is reported that CYP4A3 is a downstream target gene of PPARα, while CD36 can regulate the expression of PPARα, and are closely related to fatty acid oxidation [24,26,44,45]. Based on the evidence, we investigated whether icariin regulated lipid metabolism in the rat liver via the CD36-PPARα pathway. As we predicted, high doses of icariin up regulated the expression of CPT1α, ACOX1, MCAD, and LCAD, key regulatory proteins of downstream fatty acid oxidation, by promoting the transcription of PPARα, which further alleviated hepatic steatosis in rats.

Icariin may improve NAFLD in PCOS by increasing liver mitochondrial CD36 content. We noted that the mRNA and protein expression of CD36 was the opposite of what we expected, so we speculated whether CD36 had undergone translocation and was expressed in the presence of other organelles. It is reported that the function of CD36 is related to its subcellular localization [31,46]. Accordingly, we speculated whether CD36 underwent translocation, resulting in an opposite expression trend than expected. Firstly, to confirm our conjecture, we investigated CD36 expression in the cell membrane, cytoplasm and mitochondria. In the model group, the protein expression of CD36 in mitochondria was lower than that in the blank group, while the expression on the cell membrane was higher than that in the blank group. However, after icariin treatment, CD36 expression increased in mitochondria and decreased in the cell membrane. These results may prove that CD36 has undergone mitochondrial translocation. Furthermore, according to a previous study [31], increased mitochondrial CD36 expression leads to more acyl-CoA oxidation by mitochondria in hepatocytes, which contributes to the reduction of lipid accumulation in non-alcoholic fatty liver disease. Also, our study showed that the expression of fatty acid oxidation-related molecules downstream of CD36 were altered after icariin treatment. In conclusion, this study shows that increased expression of CD36 mitochondrial translocation can regulate downstream PPARα and then up regulate the expression of PPARα target genes, such as CYP4A3, CPT1α, ACOX1, MCAD, and LCAD, to promote hepatic lipid oxidation.

Our study confirmed that HFD and letrozole could successfully make a pathological model of NAFLD in PCOS. Icariin significantly reduced hepatic lipid deposition by increasing fatty acid oxidation and decreasing the expression of fatty acid synthesis-related genes (Figure 7). It further alleviated non-alcoholic fatty liver disease in rats with polycystic ovary syndrome and provided a new strategy for the future treatment of NAFLD in PCOS by Chinese medicine. Meanwhile, the limitation of this study is that we only demonstrated that CD36 can act through mitochondria. However, we did not dig deeper into the metabolism of fatty acids in vitro.

## 5. Conclusions

In our study, we created a model of non-alcoholic fatty liver disease in polycystic ovary syndrome using letrozole and a high-fat diet, and we investigated the pharmacological effects of icariin on non-alcoholic fatty liver disease in polycystic ovary syndrome by administering it at doses of 40 mg/kg and 80 mg/kg. Through the CD36-PPAR-CYP4A3 pathway, our work showed that large doses of icariin can greatly reduce lipid buildup in the liver and increase mitochondrial fatty acid oxidation in the liver, further reducing non-alcoholic fatty liver disease. Additionally, we have demonstrated that icariin can lessen the liver’s creation of fatty acids from scratch and lipid buildup.

## Figures and Tables

**Figure 1 molecules-28-00517-f001:**
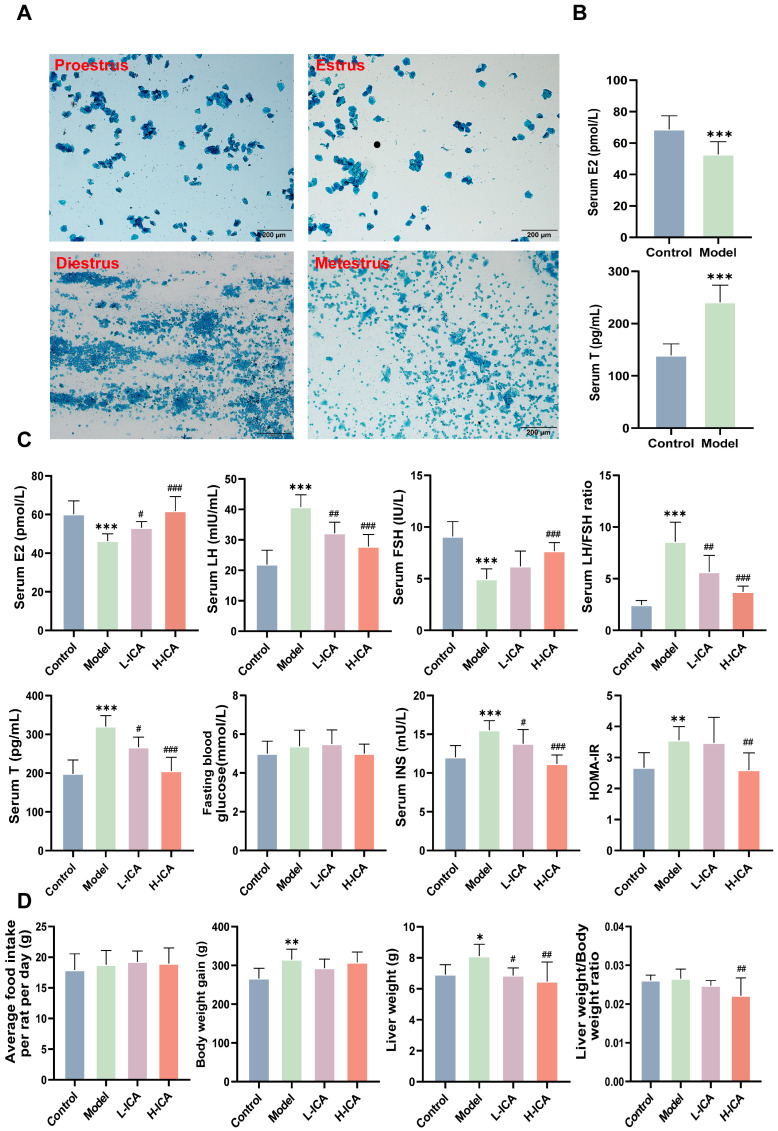
The rat model and sex hormone levels. The main types of vaginal exfoliated cells in vaginal smears of rats (scale bar = 200 µm) (**A**). Proestrus: predominantly nucleated epithelial cells, Estrus: predominantly keratinized epithelial cells, Diestrus: predominantly white blood cells, and Metestrus: Visible keratinized epithelial cells and leukocytes. levels of E2 and T were determined by ELISA (**B**). After 28 days of administration, E2, FSH, T, LH, LH/FSH ratio, INS, and HOMA-IR levels were measured by ELISA kits (**C**). Body weight gain, food intake, and liver weight of rats were collected (**D**). *n* = 36; values are mean ± SEM; Control vs. Model, * *p* < 0.05, ** *p* < 0.01, *** *p* < 0.001; Model vs. drug group, ^#^
*p* < 0.05, ^##^
*p* < 0.01, ^###^
*p* < 0.001.

**Figure 2 molecules-28-00517-f002:**
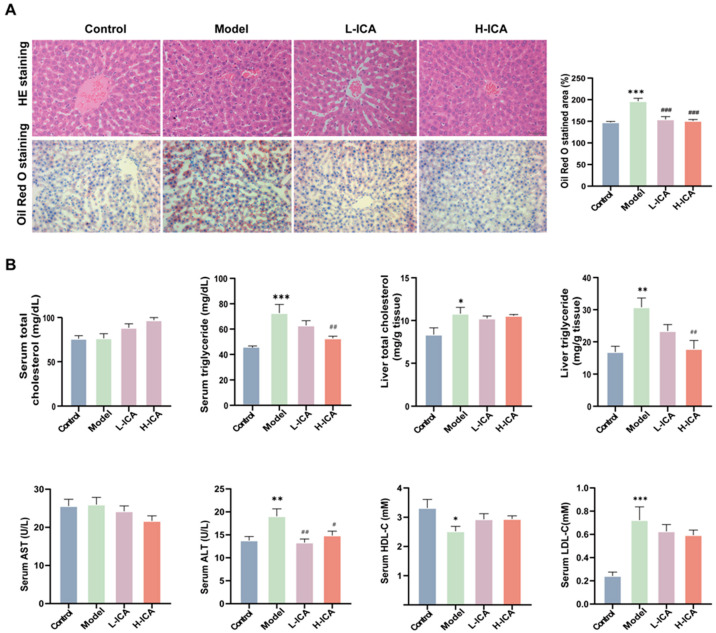
Icariin improves liver steatosis and liver function in rats. The liver sections of rats were stained with hematoxylin and eosin (scale bar = 50 μm) and Oil red O (scale bar = 200 μm) (**A**). Serum specimens were collected for TC, TG, AST, ALT, HDL-C, and LDL-C; liver samples were collected for TC and TG (**B**). *n* = 6, values are mean ± SEM; Control vs. Model, * *p* < 0.05, ** *p* < 0.01, *** *p* < 0.001; Model vs. drug group ^#^
*p* < 0.05, ^##^
*p* < 0.01, ^###^
*p* < 0.001.

**Figure 3 molecules-28-00517-f003:**
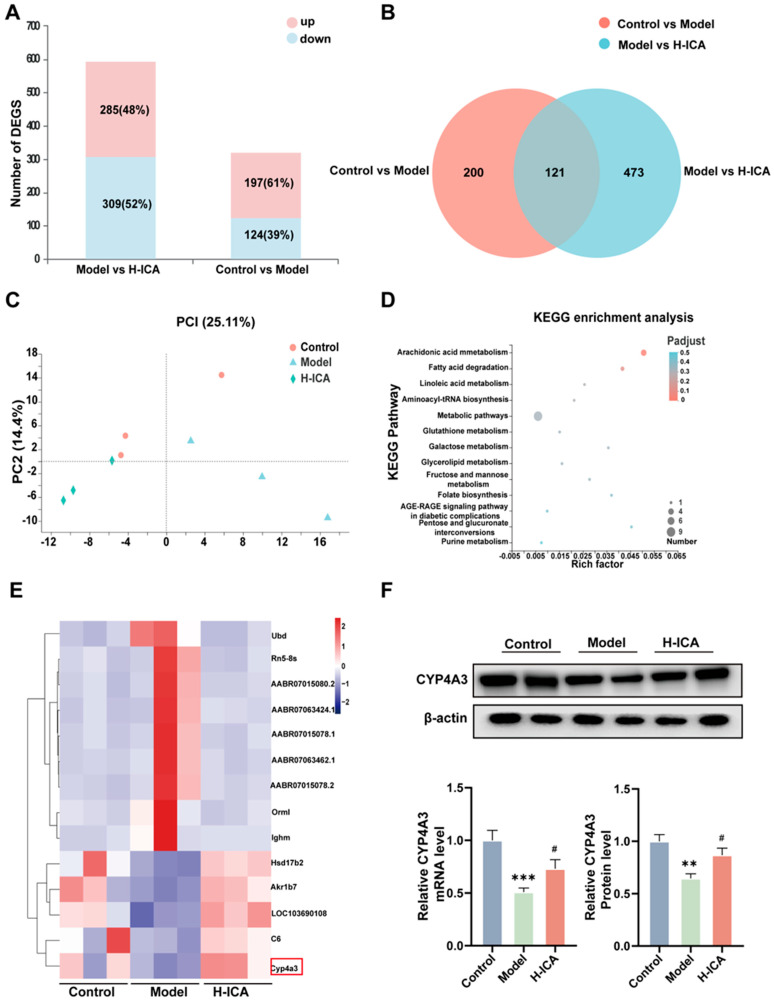
Effects of icariin on NAFLD in PCOS models as revealed by RNA-sequence analysis. RNA-sequence analysis of the molecular mechanism of icariin amelioration of NAFLD in PCOS. Superimposed graph of differently expressed genes (**A**). Wayne plot of differently expressed genes (**B**). PCA analysis based on different groups (**C**). Molecular mechanism of KEGG pathway analysis (**D**). Heat map analysis of differentially expressed genes with high expression (**E**). Trend validation of the screened differentially expressed genes CYP4A3 by qPCR (GAPDH as internal reference) and protein (β-actin as internal reference) (**F**). *n* = 6, values are mean ± SEM; Control vs. Model, ** *p* < 0.01, *** *p* < 0.001; Model vs. drug group, ^#^
*p* < 0.05.

**Figure 4 molecules-28-00517-f004:**
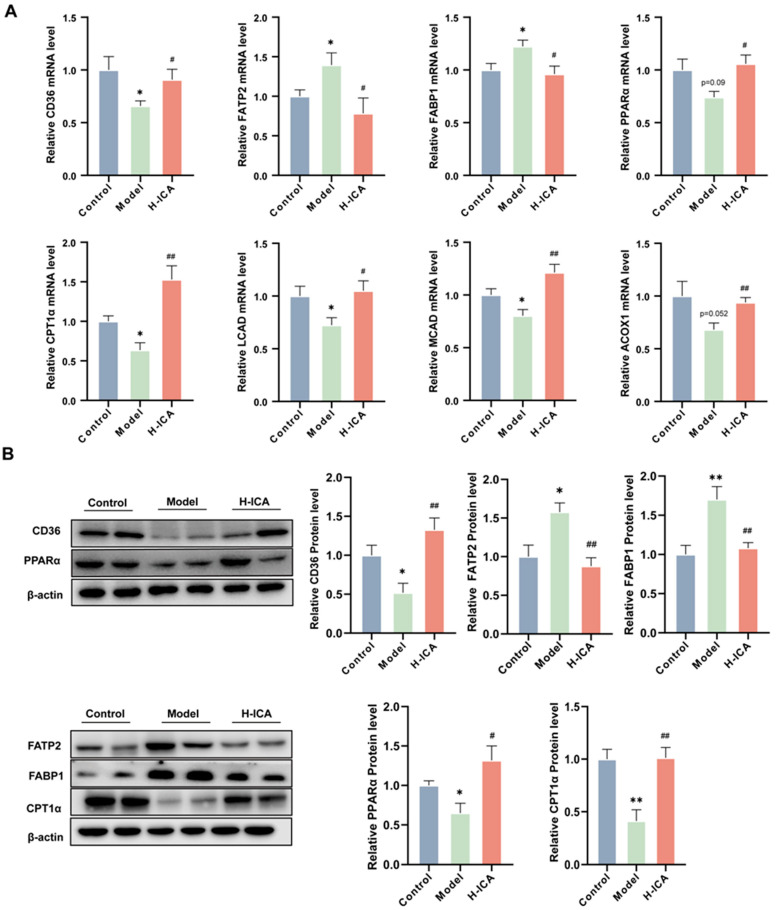
Icariin improves NAFLD in PCOS by promoting fatty acid oxidation in the liver. Gene and protein levels were verified. Gene expression (GAPDH as internal reference) (**A**); Protein expression (β-actin as internal reference) (**B**). *n* = 6, values are mean ± SEM; Control vs. Model * *p* < 0.05, ** *p* < 0.01, ^#^ (Model vs. drug group), ^#^
*p* < 0.05, ^##^
*p* < 0.01.

**Figure 5 molecules-28-00517-f005:**
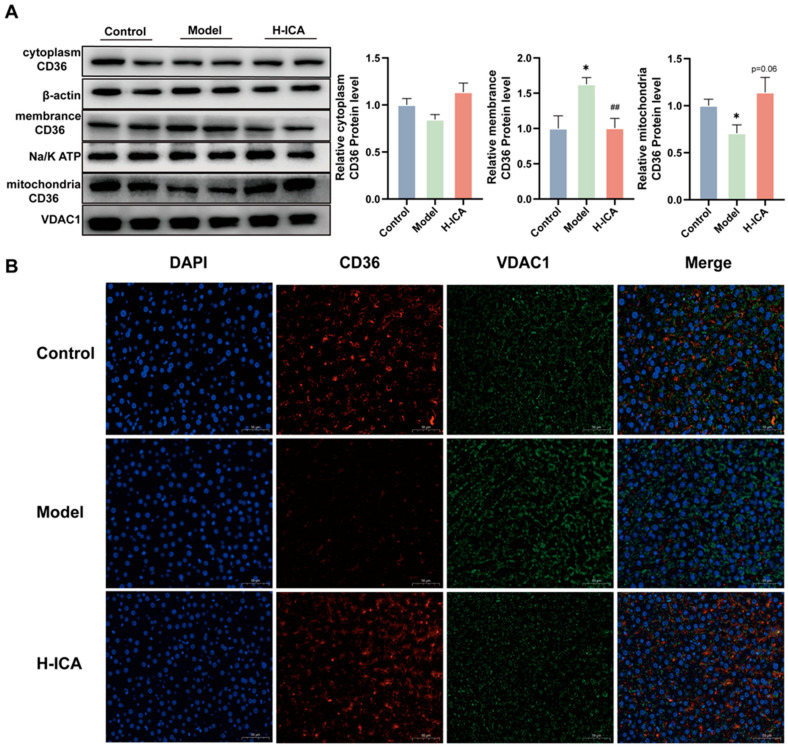
Icariin alleviates NAFLD in PCOS by increasing CD36 content in mitochondria. The protein levels of CD36 in the cytoplasm (β-actin as an internal reference), cell membrane (Na/K ATP as an internal reference) and mitochondria (VDAC1 as an internal reference) were analyzed separately. *n* = 6 (**A**). CD36 and VDAC1 double immunofluorescence staining in liver tissue (scale bar = 50 µm, *n* = 3), green fluorescence indicated the mitochondrial marker VDAC1, red fluorescence indicated CD36 (**B**). values are mean ± SEM; Control vs. Model, * *p* < 0.05, Model vs. drug group ^##^
*p* < 0.01.

**Figure 6 molecules-28-00517-f006:**
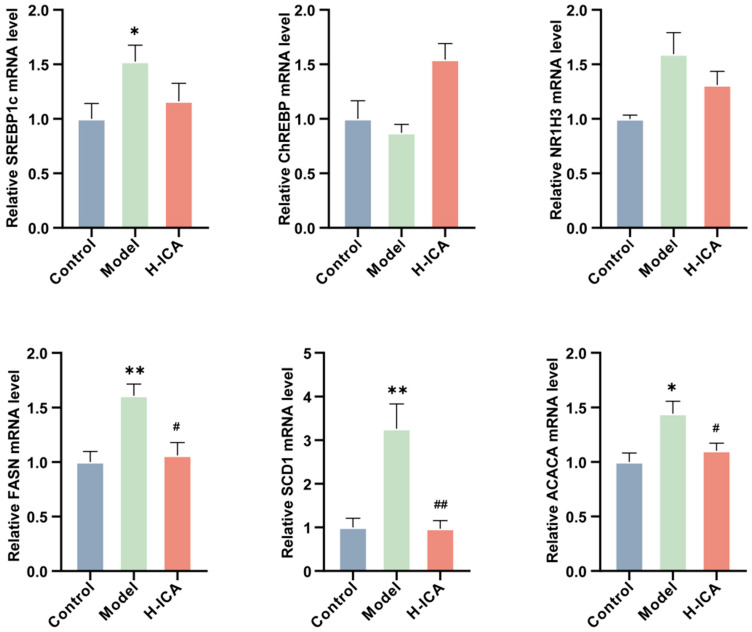
Icariin improves liver steatosis by reducing the expression of genes associated with fatty acid synthesis. Validation of genes involved in fatty acid synthesis in the liver (GAPDH as internal reference). *n* = 6, values are mean ± SEM; Control vs. Model, * *p* < 0.05, ** *p* < 0.01; Model vs. drug group ^#^
*p* < 0.05, ^##^
*p* < 0.01.

**Figure 7 molecules-28-00517-f007:**
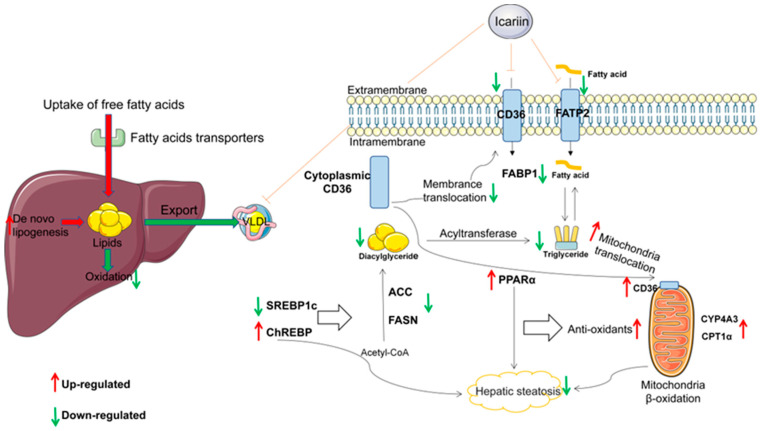
Diagrammatic representation for the mechanism of current explored study.

**Table 1 molecules-28-00517-t001:** Primers for qRT-PCR.

Gene	Forward Primer	Reverse Primer
SCD1	TGTCAAAGAGAAGGGCGGAAAGC	CAGGATGAAGCACATGAGCAGGAG
SREBP1c	CCTGCTTCTCTGGGCTCCTCTC	GCACGGACGGGTACATCTTTACAG
ChREBP	GAAGACCCAAAGACCAAGATGC	TCTGACAACAAAGCAGGAGGTG
NRIH3	GTGCCTGATGTTTCTCCTGACTCTG	AAGTGTTGCCTCCCTGGTCTCC
CD36	TGTACCTGTGAGTTGGCAAGAAGC	ACAGCCAGGACAGCACCAATAAC
FABP1	GGTCAAGGCAGTGGTTAAGATGGAG	GTAGACGATGTCACCCAGTGTCATG
PPARα	GTCATCACAGACACCCTCTCCC	TGTCCCCACATATTCGACACTC
ACACA	TTCCCATCCGCCTCTTCCTGAC	TGCTTGTCTCCATACGCCTGAAAC
FATP2	AGGTGAGGTTGGACTCTTGATTTGC	GGAGATCGCCACTGTTGAAGTAGAC
CPT1α	CAGGAGAGTGCCAGGAGGTCATAG	TGCCGAAAGAGTCAAATGGGAAGG
CYP4A3	TCTCACCAGATTCTCCTCGCCATAG	CCACAGCCACCTTCAGCTCATTC
ACOX1	AAATCAAGCAAAGCGAACCAGAACC	CGAAGTGGAAGGCATAGGCAGTG
LCAD	CCCTGGTTTCAGCCTCCATTCAG	AATACACTTGCCCGCCGTCATC
MCAD	AGAGGCTACAAGGTCCTGAGAAGTG	AACTCTTTCTGCTGCTCCGTCAAC
FASN	ACCTCATCACTAGAAGCCACCAG	GTGGTACTTGGCCTTGGGTTTA
GAPDH	TGCACCACCAACTGCTTAG	GGATGCAGGGATGATGTTC

## Data Availability

The data supporting the present findings are contained within the manuscript.

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
