# Peer review of "Icariin Alleviates Nonalcoholic Fatty Liver Disease in Polycystic Ovary Syndrome by Improving Liver Fatty Acid Oxidation and Inhibiting Lipid Accumulation"

_molecules, 2023, doi:10.3390/molecules28020517_

Round 1
Reviewer 1 Report
The manuscript presented a comprehensive study on alleviation effects of nonalcoholic fatty liver disease by Icariin. However, the effects were not strong enough as shown in Fig. 2.
For the serum total cholesterol, liver total cholesterol, serum AST, serum HDL, serum LDL, there were no statistical difference. Furthermore, for serum triglyceride and serum ALT, the high and low dose did not make much difference. So, I did not suggest to publish this manuscript.
Author Response
Thank you for your decision and comments on my manuscript. We will pay attention to your questions. Best wishes.
Reviewer 2 Report
The article entitled “Icariin alleviates nonalcoholic fatty liver disease in polycystic
ovary syndrome by inhibiting the liver fatty acid oxidation and lipid
accumulation” is written well and could be accepted in “Molecules” after some minor corrections.
1. The keywords should be arranged alphabetically in the manuscript.
2. Abstract: The abbreviation should be clarified and written.
3.Line 14: Icariin, a main component of the Chinese herb Epimedium, a number of 14
studies have shown that it alleviates abnormal lipid metabolism…The sencence should be written as….Icariin is main component of the Chinese herb Epimedium. A number of studies have shown that it alleviates abnormal lipid metabolism.
4. Line 36: An estimated 5% to 20% of women of reproductive age suffer from PCOS, an endo-
crine disorder…. Add “and” after PCOS and remove ,en.
5.Could you provide ethical certificate/propf from ethical review committee of University for the experimental study on animals/rats?
6.. Why the liver was treated with 10% formalin?
7. Figure 1: Why the standard deviations values are same in different experiments. They should be different from each other?
8.Figure 3: D remove the red rectangular portion around fatty acid metabolism?
9. Figure 7: The diagram of mechanism we explored in the current study….Rewrite as…Diagrammatic representation for the mechanism of current explored study.
10. Add some updated literature study/references 2022 in introduction section .
11. There are too much abbreviated short words in manuscript. Write their full names as well.
12. Check the plagiarism of the manuscript.
13. Conclusion is missing in the manuscript.
14.The authors should formulate 2-3 important conclusions from their conducted research work.
15. The manuscript should be revised carefully. There are some minor grammatical mistakes.
Author Response
We are grateful to reviewer #2 for his/her effort reviewing our paper and his/her positive feedback. The summary of our work as written by this reviewer is precise. Here below we address the questions and suggestions raised by the reviewer #2.
(1) We sincerely thank the reviewer for careful reading, As suggested by the reviewer, the keywords we have arranged alphabetically in the manuscript. (Line 39-40)
(2) We think this is an excellent suggestion. We have clarified and wrote the abbreviation in Abstract. Including the exact location where the change can be found in the revised manuscript. (Line 24-36)
(3) We have re-written this part according to the reviewer’s suggestion.
(4) Line 36,“and” was added and “an” was deleted.
(5) I can provide ethical certificate from ethical review committee of university for the experimental study on animals.
(6) Because 10% neutral formalin can quickly penetrate into the liver tissue, prevent autolysis and denaturation, stabilize various structures in cells, maintain the normal shape of the liver, after fixation, HE staining is bright and contrasting is obvious.
(7) I apologized that we made a mistake in the drawing process, and we have corrected the image in Figure 1.
(8) According to the reviewer’s suggestion, we have removed the red rectangular portion around fatty acid metabolism.
(9) We have re-written this part according to the reviewer’s suggestion. (Line 463)
(10) We sincerely appreciate the value comments. We have checked the literature carefully and added more update references 2022 in introduction section (Reference 4,6,10,13).
(11) Considering the reviewer’s suggestion, we have written full names of abbreviated short words.
(12) Thanks to the reviewers' careful reading of the manuscript and pertinent suggestions, I have checked the plagiarism again and rewritten.
(13) We were really sorry for our careless mistakes. Thank you for your reminder. I have added a conclusion after the discussion part of the article. (Line 464-472)
(14) Thanks for the suggestion, we have summarized 3 necessary conclusions in our conclusion section. (Line 464-472)
(15) Thanks for your careful checks. We are sorry for our carelessness. Based on your comments, we have made the corrections to the grammatical harmonized within the whole manuscript.
We tried our best to improve the manuscript and made some changes marked in blue and red in revised paper which will not influence the content and framework of the paper. We appreciate for reviewers’ warm work earnestly, and hope the correction will meet with approval. Once again, thank you very much for your comments and suggestions.
Special thanks to you for your good comments

Reviewer 3 Report
Title: Icariin alleviates nonalcoholic fatty liver disease in polycystic 2 ovary syndrome by inhibiting the liver fatty acid oxidation and 3 lipid accumulation.
Overall: the manuscript is written well and describes the hypocholesterolemic potential of Icariin in a PCOS-induced rat model; the results were supported with well-established biochemical, molecular, and histopathological techniques. I recommend “minor revision”; here are some points to be addressed:
Abstract:
The full abbreviation of words should be mentioned at their first use.
Materials and methods:
Why the model group used “letrozole 1 mg/ kg”? Explain its role in the introduction.
The authors should describe the components of a high-fat diet (D12492)
The section “2.2. Drugs and reagents” should precede the section of “2.1. Experimental animals”.
It would be better to add a flowchart describing the different groups.
Discussion:
The paragraph on “Icariin” on page 16 (lines: 367-373) should be placed in the introduction section.
The authors did not show the difference between the two doses of Icariin; therefore, I recommend using the percentage of change to the model group to demonstrate which dose exhibited more effective therapeutic potential.
Author Response
We feel great thanks for your professional review work on our article. As you are concerned, there are several problems that need to be addressed. According to your nice suggestions, we have made extensive corrections to our previous draft, the detailed corrections are listed below.
Abstract:
We think this is an excellent suggestion. We have clarified and wrote the abbreviation in Abstract. Including the exact location where the change can be found in the revised manuscript.
Materials and methods:
(1) Letrozole is an aromatase inhibitor that is reported to affect testosterone accumulation and upregulate liver triglyceride levels after 21 days of administration. Therefore, we used 1 mg/kg of letrozole and HFD co-administration to construct the rats model. I have explain its role in the introduction. (Line 93-96)
(2) Thanks for your advice, I have supplemented the main components of a high-fat diet (D12492) in materials and methods. (Line 111-113)
(3) We think this is an excellent suggestion. We have preceded the section “2.2. Drugs and reagents” into the section “2.1. Experimental animals”. (Line 128-140)
(4) We have added a flowchart describing the different groups according to the reviewer’s suggestion.
Discussion:
(1) We sincerely thank the reviewer for careful reading. As suggested by the reviewer, we placed the paragraph on “Icariin” on page 16 (lines: 367-373) in the introduction section.
(2) Thank you for the advice, this suggestion is of great help to our article. We compared treatments with different doses of medication. The supplemented content is described in "3.1Rat model and sex hormone levels" and "3.2. Icariin improves liver steatosis and liver function in rats”. (Line 245-253, Line 273-275)
Special thanks to you.
Other changes:
The title “Icariin alleviates nonalcoholic fatty liver disease in polycystic ovary syndrome by inhibiting liver fatty acid oxidation and lipid accumulation” was changed into “Icariin alleviates nonalcoholic fatty liver disease in polycystic ovary syndrome by improving liver fatty acid oxidation and inhibiting lipid accumulation”
I would like to take this great opportunity to thank you and the reviewers for the valuable comments. We have addressed all the comments carefully, and the revised portions are highlighted in blue and red in the manuscript. I believe that the addressing of these comments has greatly improved the quality of this manuscript.
Round 2
Reviewer 1 Report
Through revision, the manuscript has been improved. Though the bioactivity is not very strong, the work is quite comprehensive. I suggest acceptance.
Author Response
Thank you very much for your suggestions on this article and we will continue to do more in depth and extensive research on this drug to evaluate its potential basic and clinical applications.